# The Effect of Perceptions on Tourism: An Econometric Analysis of the Impacts and Opportunities for Economic and Financial Development in Albania and Kosovo

**Mirela Tase** [1] and **Enkeleda Lulaj** [2,*]

1 Department of Tourism, University "Aleksander Moisiu", Currila, 2000 Durres, Albania; mirelatase@uamd.edu.al
2 Faculty of Management in Tourism, Hospitality and Environment, "Haxhi Zeka" University, Eliot Engel, 30000 Peja, Kosovo
* Correspondence: enkeleda.lulaj@unhz.eu

**Abstract:** Today, tourism plays an important role in the economic and financial development of countries, and its impact is greater than ever. Therefore, for sustainable economic and financial growth and well-planned development, public and private investments should be directed to areas of priority tourism development. Research on the effect of perceptions on the behavior of tourists in these two countries has not been carried out before, thus, the purpose of this research is to determine whether the effects of the perception of tourists has an impact on economic and financial development based on factors (F1 (f.1.1 and f.1.2) and F2 (f.2.1 and f. 2.2)). For this study, the data were provided by respondents from several cities in Albania and Kosovo. A total of 1002 questionnaires divided into three sessions were analyzed using factor analysis, data reliability analysis, and multiple regression analysis. All analyses were performed using SPSS version 23.0 for Windows. In this case, 23 variables tested and divided into two factors and five sub-factors. The results showed that special attention should be paid to the following factors: (a) awareness of tourists of facilities in tourist destinations where they would like to visit; (b) greater knowledge of foreign languages for residents of both countries, which could facilitate communication with tourists during purchases or other requests; (c) attracting new investors and the creation by the government bodies of conditions and security for both investors and tourists; (d) supporting the marketing and sale of local products for tourists; (e) the need for infrastructure support from government bodies in both countries in order to increase economic and financial well-being through tourism; (f) the need to implement strategies focusing on the sustainable development of both countries through tourism should be strengthened; (g) in terms of sustainable development and regional competitiveness, Kosovo and Albania should follow development trends and be competitive with other countries in the region. The implications of this paper relate only to certain studied variables, and only in certain cities of Albania and Kosovo. In case of future analysis by different researchers, other variables can be analyzed for different locales by making comparisons with the presented data.

**Keywords:** tourism; economy; finance; tourist behavior; perceptions; econometric models; Albania; Kosovo; development

## 1. Introduction

Year after year, tourism is increasingly crystallizing as one of the main engines of the country's economic development. The tourism industry occupies a key place in the economy and is an important source for the development of Albania and Kosovo. This study aims to analyze the effect of perception on the behavior of tourists in these two countries, which has not been studied before. The economic, financial, and social systems of most countries have been disrupted during the last several years due to the COVID-19 pandemic, which makes it very difficult to assess short- and long-term consequences [1]. Globally,

travel and tourism are significant contributors to leading sector for job creation and socio-economic and cultural development [2]. Measures and preparation strategies such as staying at home, keeping a distance of 1.5 m, community lockdowns, and measures against crowding have stopped tourism and leisure in global travel [3]. The tourism industry is particularly vulnerable to crises or disasters [4]. In Albania and Kosovo, 'lockdown' measures started in March 2020 and tourism and hospitality businesses were closed. Understanding how a destination responds to a tourism crisis from a supply-side perspective, especially in the two countries that are the subject of this study, is very important and timely, as it has implications for other developing countries [5]. These countries are located in Southeastern Europe, and tourism and travel are a significant part of their national economies. Albania is included in the list of countries with great natural, historical, and cultural heritage potential (Figure 1). While Albania offers all types of tourism, blue tourism holds the main weight in the country's GDP. The advantage is that tourism in Albania benefits from a large diversity of beautiful landscapes in a relatively small space, offering many possibilities for various activities all year round and in all seasons, although primarily in summer and winter. Natural and rural areas in Albania offer opportunities for the development of rural tourism, mountain tourism, ecotourism, and outdoor activities (rafting, parachuting, mountain biking, fishing, trekking, mountaineering, hiking, horseback riding, study tours, etcetera). In Albania, the direct contribution of tourism to GDP in 2018 was 8.5%, and it is projected that by 2028 the direct and indirect effect of the tourism sector will reach approximately one third of Albania's total GDP. For this industry to become one of the pillars of the Albanian economy, the seasonal effects of coastal tourism need to be mitigated by developing other forms of tourism, increasing the number of visitors and overnight stays, and consequently tourism revenue.

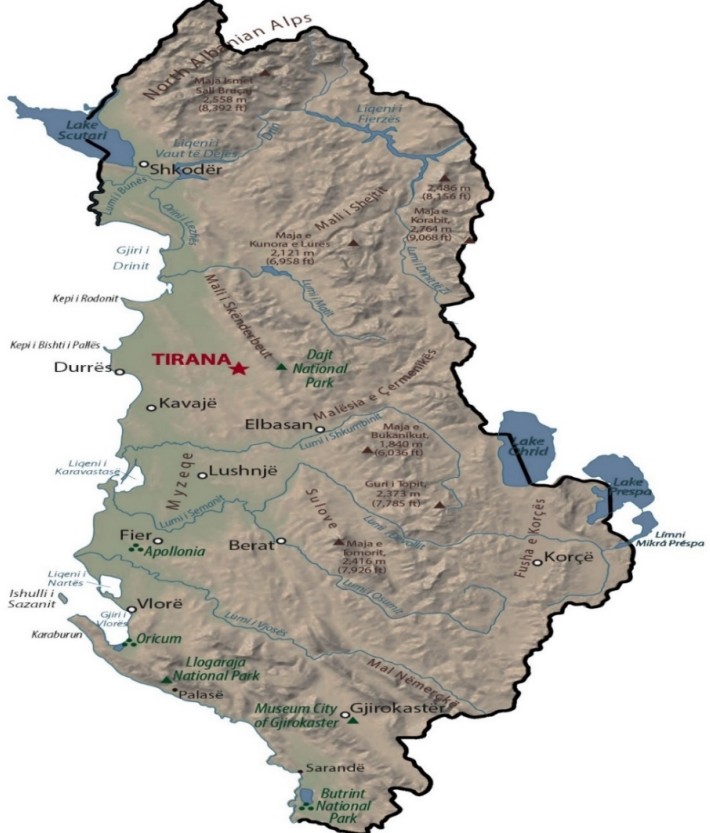

**Figure 1.** Physical Map of Albania; jpeg prepared by Mirela Tase.

As a mountainous country, Kosovo is landlocked within its mountains and has to compete both with her neighbors and with other destinations (Figure 2).

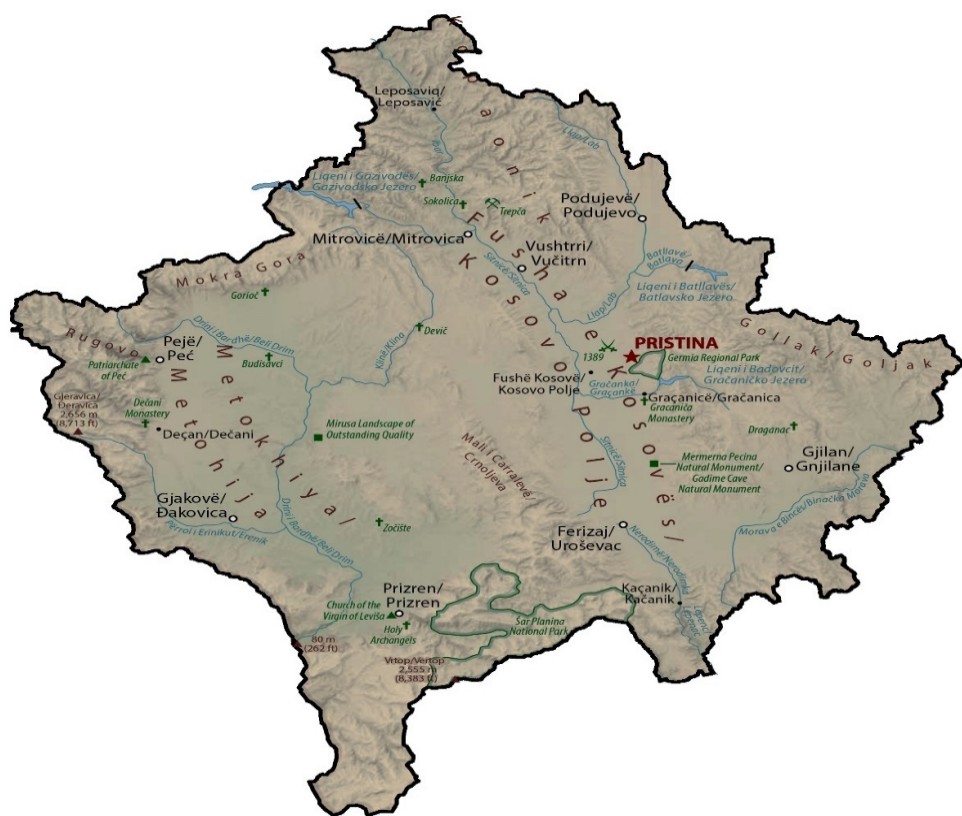

**Figure 2.** Physical map of Kosovo; jpeg prepared by Mirela Tase.

Kosovo has no direct access to the sea and possesses a mostly hilly and mountainous terrain; however, it has a favorable climate and multiple natural, cultural, and historical resources, which constitute a resource basis for tourism [6]. However, compared to other Balkan countries, the development of tourism in Albania and Kosovo is far from the potential represented by these countries' natural, historical, and cultural assets. Infrastructure, quality of services, andtourism offerings and products are all factors that have somewhat inhibited the sustainable and consistent development of tourism in both countries. When promoting destinations in Albania and Kosovo, stakeholders and managers must consider aspects related to sustainability, and must understand how tourists perceive the destination and what kind impact they have in it. The tourism and travel industry are influenced by a variety of internal and external factors, such as political instability, economic conditions, environmental conditions, etc. [7]. According to the UNWTO, the tourism industry has been one of the sectors most stricken by the COVID-19 pandemic, resulting in a loss of up to USD 50 billion in spending and a decline in international tourist arrivals by up to 3% worldwide [8]. The tourism industry is considered the major contributor to GDP for several countries, and as of 2020 over 100 million direct tourism jobs were at risk [9]. The tourism and leisure industry plays an essential role in the economic activities of both studied countries, and customer satisfaction has an important impact on the industry [10]. In popular destinations such as Paris or London, over-tourism has been created [11]. Cities such as Tirana, Prishtina, Durres, and Peja are turning into highly sought-after destinations for international tourists thanks to the attractions they offer. Tourism and travel provide an important contribution to business operations, and ultimately contribute to the worldwide economy. The travel and tourism sector has led to significant growth in the GDP of both countries, mainly due to the strong impact of tourism and transport [12]. Based on estimated 2018–2019 data on the tourism industry in Albania and Kosovo, the countries showed a combined contribution to GDP of USD 5.0845 million dollars. According to the WTTC, the contribution of tourism to GDP in Albania was 8.4% in 2016, while in 2019 it was 21.2%, or USD 3.264.5 million [13]. In Kosovo, the contribution of tourism to GDP in

2019 was 9.3%, or USD 1.820.0 million [14]. The present study reports the findings of both quantitative and qualitative interviews. We engaged in over one thousand interviews with different target groups. This quantitative study examines the impact of tourists' perceptions. The interviews were conducted in the cities of Tirana, Durres, Fier, Pristina, Peja, Gjakova, and Istog, which have both the largest population concentrations and the largest tourist flows. Tourism has the potential to generate even more jobs in the future; however, there is a need for education and training of tourism sector workers in order for the service provided to visitors to be professional [14]. This paper therefore examines the perceptions of tourists who have greater preference for small regions, cities, and places where they can have unique experiences. Perceptions associated with shopping for local products and quality of service while on vacation are likely to impact tourists' shopping behavior. The effect of perceptions on tourism in Albania (Al) and Kosovo (Ko) has not previously been analyzed through econometric models in terms of the impacts and opportunities with respect to economic and financial development. In the first stage, this research study investigates the significance and settings of the COVID-19 pandemic [15]. In the second stage, we focus on creating a model for econometric analysis of the effects of perception on tourism and opportunities for economic and financial development in Albania and Kosovo. Due to the relevance of the tourism sector in the economy and financial developement of both countries, it is nessesary to promote a sustainable developement system with tourism as a powerful sector that promotes economic growth and facilitates the creation of sustainable and inclusive employment [14]. Therefore, this research aims to understand the effect of perceptions on tourist behavior and the impact on economic and financial development based on three main issues: (a) a new approach to the facts around tourists' perceptions in both countries; (b) analysis of the impact of tourism on economic and financial development; (c) discovering new ways to use these opportunities to improve economic and financial development in both countries through tourism.

## 2. Literature Review

The tourism industry of is considered a pillar of the economy in developing countries and plays a major role in their GDP. Tourism can generate significant revenues, especially with the creation of services and activities that extend touristic offerings and lead to greater direct expenditures by tourists. Tourism, as a critical sector of local and national socio-economic development, relies heavily on energy use. Tourism has been the hardest-hit sector the COVID 19 pandemic, which started in mid-March of 2020, blocked all trips for several months; even afterwards, they remained limited, and continue to be [13].

The COVID-19 pandemic has resulted in global challenges related to renewable energy, carbon emissions, and economic and healthcare crises, and has had spillover impacts on global industries, including tourism and travel, that are a major contributor to the service industry worldwide [16–18]. Any rise in the number of tourist arrivals requires an increase in energy demand to support the change [19]. Many countries and regions that are tourist destinations around the world wish to analyze and improve their tourism dynamics and increase tourism's economic contributions. According to [20], tourism contributes to national economic growth and development and improves the standard of living, thus promoting a process of regional convergence and stimulating domestic demand. This study is focused on analyzing the impact of tourism on economic growth in Albania and Kosovo and the effect of perceptions on tourist behavior [21].

The tourism industries in Albania and Kosovo needed to begin positioning themselves as one singular destination that invites tourists for exploration and adventure. In our view, Albania and Kosovo should find mechanisms to increase demand and create facilities for services, especially transport and health care, as drawn from the perceptions of our respondents. Facing a crisis is not uncommon for companies in the tourism industry such as travel agencies, hotels, and transport, as almost every tourism company is faced with extraordinary events over time [22]; however, the occurrence of tourist crises often leads to a loss of faith and confidence in a destination's safety. Perhaps one of the most pressing

problems in tourism operations today relates to how businesses across all sub-sectors of the industry will be able to maintain the confidence and physical security of both customers and employees [23].

The governments of Albania and Kosovo must increase their investment in developing such programs as well as in public education to deliver them in order to enhance social resilience and the sense of security felt by both tourists and communities, as this can have a substantial impact on the travel decisions of individual travelers and on travel behavior as a whole [24]. Fishbein and Ajzen (1975) emphasize that a person's intention with respect to current issues in tourism is determined by his/her attitude and by individual perceptions [25]. The focus of their research is on addressing questions such as whether host communities should remodel their tourism offerings in order to comply with the changing demands of tourists. Even though this theory is used in different disciplines, our paper is focused on the tourism sector, where this theory has been used to explain the decision-making process for destinations [26] and the responsible behavior of tourists [27]. In F1.1., the variable that has the highest value is Q4.2 = 0.771 (tourists appreciate the country's cultures and lifestyle), while in F1.2., the variable that has the highest value is Q4.5 = 0.824; both of these variables have great importance for economic and financial development. As a part of the European region, blue tourism in Albania and mountain tourism in Kosovo are considered strategic priorities as instruments for development in specific regions of both countries. This can be approached as an integrated program in which, directly or indirectly, all sectors of society and the economy contribute. For Albania's Ministry of Tourism, economic, environmental, and socio-cultural sustainability are prerequisites for the development of the tourism sector. Although facing more problems in different stages of development, tourism development in the Republic of Kosovo is already moving in the same direction as the overall development of the country's economy. Kosovo, through placing itself in the center of the Balkan Peninsula as a tourist destination, is an important area that can be involved in the development of tourism both in the region and in Europe [28].

Tourism and hospitality are among the most important activities for the economic development of Kosovo. Regardless, the travel and tourism industry has developed positive impacts and is a significant contributor to the economy of both countries. Albania is increasingly recommended as a tourist destination by various tour operators and international travel guides. In terms of their contribution to tourism revenue, we can classify tourism products in Albania into three main categories. These categories consist of the forms of tourism development, which occupy an important role in the sector's contribution to the overall economy. The main products can be classified as follows: (a) coastal tourism; (b) natural tourism; and (c) thematic tourism. The contribution of tourism to GDP in Albania, according to the 2018 Economic Impact Report of the World Travel and Tourism Council1 (WTTC), in 2017 the tourism sector recorded a direct contribution of USD 1.12 bn, accounting for about 8.5% of the Gross Domestic Product (GDP). The regions in Kosovo most visited by international guests in 2015 were Pristina (53,057), Peja (12,694), and Prizren (9779) [6]. Regarding perceptions of state support for tourism, as both countries are known for their tourism, according to the findings and recommendations in [29] it is emphasized that a fair budget allocation for all areas must be made according to the needs of the country.

The contribution of tourism to GDP in Kosovo in 2019 was 9.3% [14]. Overall [9], the number of foreign tourists fell by 29%, a much larger decrease than the previous global decrease of 3.99% which occurred in 2009. Kosovo is emerging as a tourist destination, and it is ideal for a relaxing long weekend or an excursion as a part of a longer tour including the neighboring countries of Albania, North Macedonia, and Montenegro. Most tourists in Kosovo, almost 79%, visit these countries as well. Another view relating the number of tourists and their contribution to the economy according can be found in [9], who studied countries such as Germany and Austria, where tourism is not a major pillar of the economy. They estimated that tourism contributed USD 3780.55 billion to Germany's

GDP and USD 446.31 billion to Austria's GDP. These two countries represent the highest human development standards as indicated by the social and economic dimensions. This means that Albania and Kosovo should increase their tourism industry based on these two dimensions because tourism has a huge impact on both GDP and on the well-being of the inhabitants.

The world has experienced a number of major epidemics/pandemics in the last 40 years, yet none has had the same implications for the global economy as the COVID-19 pandemic. While not as contagious as measles and not as likely to kill an infected person as Ebola, people can start shedding the virus several days in advance of symptoms [30,31]. As a result of travel restrictions and lockdowns, global tourism has slowed down significantly, with the number of global flights dropping by more than half, and travel bans have grounded a growing number of carriers. Passenger numbers are likely to have declined even more steeply; many airlines have adopted specific seating policies to maintain a distance between customers [32]. In a travel context, subjectively perceived risk can affect tourists' destination choices and travel behavior [33]. While in F1.2. the variable has the value Q4.7 = 0.648, tourists respect the rules and regulations of the host country and have great importance in the economic and financial development in Albania and Kosovo. This finding can be rationalized by its feasibility, low cost, and widespread acceptance even before the pandemic. The respondents' level of agreement with selective restrictions on foreigners from high-risk countries was expected. Restrictions on travel and tourism activities, which are ongoing, have made travel agencies more affected by the pandemic in the country.

This sector is failing to recover. Restrictive measures have been in force since 2020, which is why Kosovo has lost the largest number of tourists. In 2020, there will be no growth. In the future, economic developments are expected to return to normal, depending on the duration of repentance. The relaxation of travel restrictions has contributed to easing travel restrictions, lifting consumer confidence and gradually restoring safe mobility in Europe and other parts of the world [34]. This may provide an impetus for individuals to transform their travel behaviors; however, a transformation of the tourism system is extremely difficult [35]. In order to attract tourists and increase economic and financial development according to [36] reforms should be made for the preservation of public money and the fair distribution of expenditures, as well as the provision of funds at both central and local levels. Following the research in [37] on the connection between tourism and economic growth, there has been further study on this relationship, and it continues to generate significant interest [38]. According to our findings, the relationship between tourists and residents is of great importance in the economic and financial development of Albania and Kosovo. Tourism's positive outcomes in a community include cultural exchange and economic benefits, a higher employment rate, increased economic activity, more advanced infrastructure for commerce, and a higher quality of life [39,40]. Based on our findings, the perceptions of tourists regarding respect for the cultural values of host communities are of great importance in economic and financial development in Albania and Kosovo. This suggests that local residents should be encouraged to visit local attractions because of their awareness of the local epidemic situation, and thus might be more confident engaging in tourism-related activities with family in local areas [41].

This research on the effect of perceptions in tourism through econometric models regarding the impacts and opportunities of economic and financial development in the two countries of Albania (AL) and Kosovo (KO) has not been conducted before; thus, the purpose of this scientific research is to show the effects of perceptions on the behavior of tourists and the impact of tourism in both countries, which are the impacts and opportunities for economic and financial development in both countries through tourism. The distances between the interviewees and the variables are in linear agreement and within a relation. In this context, the characteristics of the two countries in terms of economic and financial development through tourism during the COVID-19 pandemic are determined based on the results of the data in the tables of both factors for the three groups. Thus,

in our findings, security and development are of great importance to the economic and financial development of Albania and Kosovo.

## 3. Materials and Methods

### 3.1. The Purpose of This Research

This research examines the effect of perceptions in tourism through econometric models regarding the impacts and opportunities of economic and financial development in two countries, Albania (AL) and Kosovo (KO), where similar research has not been conducted before. Thus, the purpose of this research is to show the effects of perceptions on the behavior of tourists and the impact of tourism in both countries. The objectives of the present paper are to determine the impacts and opportunities of economic and financial development in both countries through tourism based on factors F1 (sub-factors F1.1 and F1.2) and F2 (sub-factors F2.1 and F2.2), the positive economic and financial impact from tourists in Albania and Kosovo, the positive impact on tourists' compliance with rules tourists, the opportunities for economic and financial development in Albania and Kosovo, the economic and financial improvement through tourism in Albania and Kosovo, as well as the growth and well-being of families through tourism in Albania and Kosovo. Through this research, we attempt to confirmed our hypotheses based on the factors taken into account in this study. Therefore, this research (a) brings a new approach to the facts concerning perceptions of tourissts in both countries; (b) analyzes the impact of tourism on economic and financial development; and (c) provides novel information about how to use opportunities in both countries to improve and enhance economic and financial development through tourism. As mentioned in the introduction, the questionnaire used in this study was completed using Likert scales (from 1—totally disagree to 5—to totally agree).

### 3.2. Methods

In this research, data were collected from Albania and Kosovo through a survey conducted with their citizens in several of the main cities in both countries (Tirana, Durres, Fier, Pristina, Peja, Gjakova, and Istog). While we met the residents of these cities, the questionnaire was completed by sending it online in the form of a link along with a request to the participants advising them of their rights; at the beginning of the questionnaire, we noted that all data are confidential and used only for research purposes. In this case, all respondents expressed their willingness to complete the questionnaire and provide their opinions on the effects of perceptions of tourist behavior and the impact of tourism on the economic and financial development of both countries. As mentioned in the literature review and introduction, cities such as Tirana, Prishtina, Durres, and Peja are turning into highly sought-after destinations by international tourists for the attractions they offer while making a significant contribution to business operations and economic development, while Fier and Istog are two cities that are trying to develop tourism and tourist attractions. However, this does not mean that other research should not include other cities; our goal was to explore these places because of the great importance of tourism. Tirana is the capital of Albania, while Prishtina is the capital of Kosovo. Peja is known for tourism and mountain tourist attractions, and Durres is known for tourism and maritime tourist attractions. There are a large number of tourists in these areas, thus, we conducted this research to investigate the effect of their perceptions. In order for this paper to be as reliable and as qualitative as possible, methods and tests have been used that fit the questions constructed within the questionnaire according to the Likert scales based on the research of other authors who have used the same methods. Factor analysis is appropriate to derive several factors from a number of variables before being used in other analyses such as multiple regression analysis. It helps to develop a questionnaire where there is no clear understanding of the variables by influencing irrelevant questions to be extracted from the model. This analysis helps our research to know exactly the factors related to the effects of perceptions in tourism on which factors to focus and not on a large number of

parameters [42]. According to [43], factor analysis is included in IBM SPSS Statistics as a data reduction technique using three main steps: to assess the suitability of the data, to derive the factors, and to satisfy the Kaiser criterion. The Kaiser criterion of sampling suitability has been studied for several levels each of p, the number of variables, and q, the number of factors within the research [44]. According to [45], factor analysis was used to analyze the impact of tourism on Nepal's economic development. We calculated the Kaiser–Meyer–Olkin criterion to check the suitability of the sample and the Bartlett Sphericity Test to assess the suitability of the data [46]. Multiple regression analysis is a static technique which is used to analyze the relationships between the dependent variable and several independent variables [47]. Multiple regression emphasizes an extension of linear regression and is part of the general static linear family such as analysis of variance, covariance, *t*-test, etc. [48]. The questionnaire was divided into three sessions, which were analyzed through three analyses: factor analysis, reliability analysis, regression analysis, and eleven econometric tests (KMO Variance, ANOVA with Tukey's Test, H telling's, Friedman T, Cronbach's Alpha, Split-Half Coefficient, Split-half model, Spearman–Brown Coefficient, Guttman Model, Lambda, Parallel Model, and Strict Parallel Model) using SPSS version 23.0 for Windows. The analysis involved several processes where certain factors were deleted in order for the model to become acceptable; as stated in the conceptual model, the data are reliable if KMO is above 0.600, Alpha is above 0.600, and R square is above 0.800. In this case, 23 variables were tested, divided into two factors and five subfactors. Reliability refers to the durability of a measure [49]. Descriptive statistics were used to describe the characteristics of the sample in terms of age, gender, education, income, country, etc. [42].

### *3.3. Data Analysis*

Research data on the effect of perceptions in tourism through econometric models regarding the impacts and opportunities of economic and financial development in two countries, Albania (AL) and Kosovo (KO), were analyzed through the three previously-mentioned analytical methods.

### *3.4. Hypotheses*

**H$_0$.** *There is no difference between the importance of variables according to residents' perceptions regarding the impact on economic and financial development through tourism in AL & KO.*

**H$_{1A}$.** *There is a difference between the importance of variables according to residents' perceptions regarding the impact and economic-financial development through tourism in AL & KO.*

AL and KO = β0 + β1 (Perceptions about the behavior of tourists on increasing economic and financial development) + β2 (perceptions on the impact of tourism on economic and financial development) + μ

or

$$H_0 = β1 = β2 = 0$$

$$H_{1A} = β1 \neq 0—\text{not all parameters are equal to zero.}$$

## 4. Results

The results included the findings for the two factors, F1 and F2, and their subfactors (F1.1 and F1.2 and F2.1, F2.2, F2.3) as follows:

### *4.1. Results for Factor I-Perceptions of AL and KO Residents on Tourist Behavior (Economic and Financial Analysis)*

Factor I (F1) or Effects of perceptions in AL and KO on tourist behavior (Figure 3) (impact analysis and opportunities for economic and financial development) includes two sub-factors, as follows:

-   F1.1 = Tourism adapts, evaluates, and visits AL and KO (the positive impact and opportunities in economic and financial development)
-   F1.2 = Tourists respect the tourist rules in AL and KO (positive impact and opportunities in economic and financial developmet)

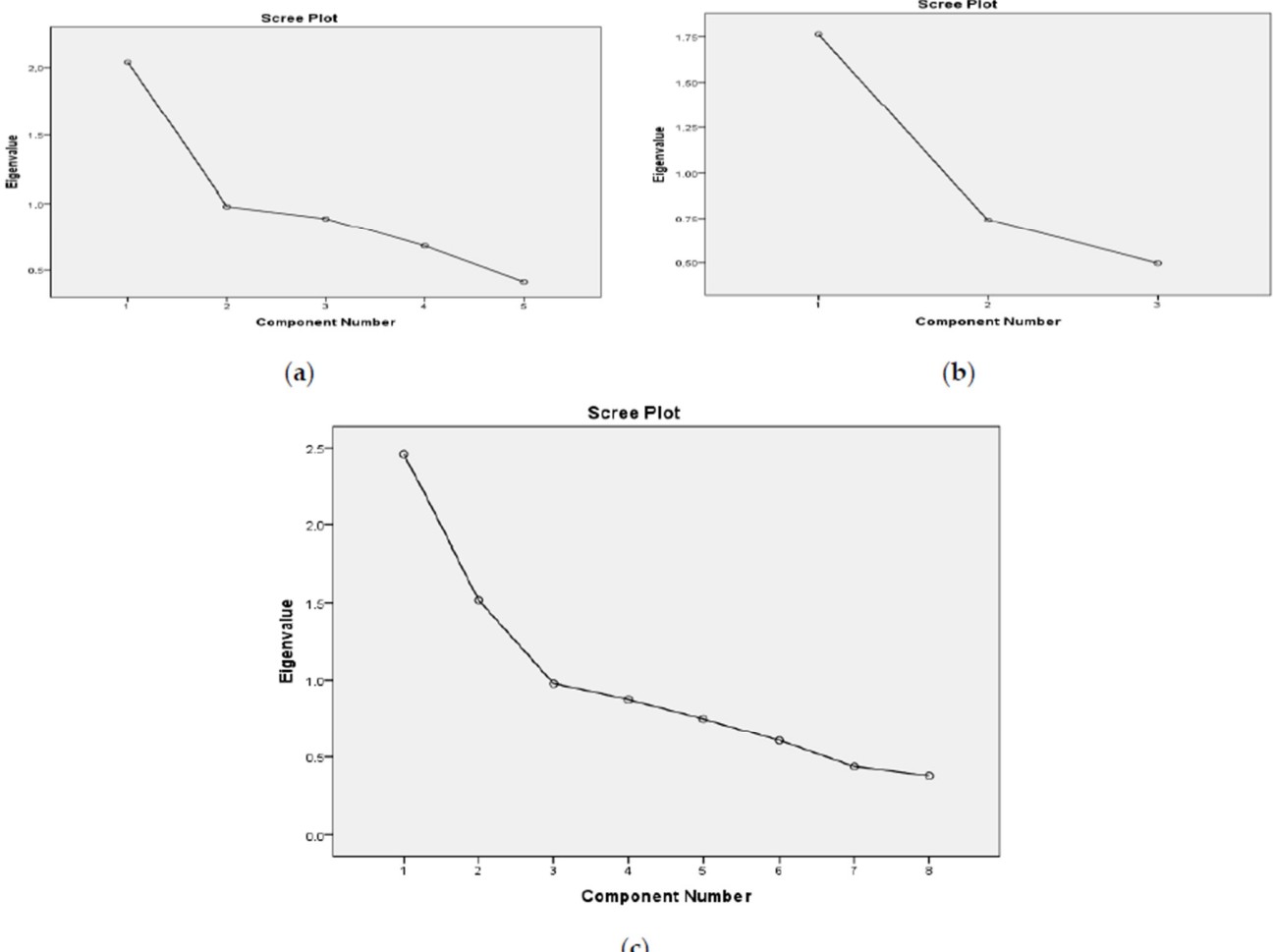

**Figure 3.** (**a–c**) Factor I (sub-factors 1 and 2) Economic–financial analysis in AL and KO through residents' perceptions of tourist behavior.

Table 1 presents descriptive analyzes for the variables age and gender per 1002 participants. The maximum is 1 while the maximum is 3.

**Table 1.** Descriptive statistics.

| Descriptive Statistics | | | | | | |
|---|---|---|---|---|---|---|
| | **N** | **Minimum** | **Maximum** | **Mean** | **Std. Deviation** | **Variance** |
| Place | 0 | | | | | |
| Sex | 1002 | 1.00 | 3.00 | 1.8333 | 0.44684 | 0.200 |
| Age group | 1002 | 1.00 | 3.00 | 1.2451 | 0.51581 | 0.266 |
| Valid N (listwise) | 0 | | | | | |

Table 2 presents the results for the importance of the eight variables from the Rotated Component Matrix to the factor analysis for F1 and subfactors F1.1 and F1.2 (0.771; 0.081; 0.684; 0.177; 0.640; 0.267; 0.514; −0.283; 0.495; 0.003; −0.045; 0.824; 0.032; 0.745; 0.387; 0.648) as well as the results from the Friedman test rankings (5.04; 4.95; 4.24; 3.89; 5.53; 4.04; 4.14;

4.16). According to the matrix in the factor analysis for both subfactors (F1.1 and F1.2), it is emphasized that the variables which have an effect are F1.1 = Q4.2; Q4.1; Q4.3; Q4.6, while F1.2 = Q4.5; Q4.8; Q4.7. In F1.1, the variable that has the highest value is Q4.2 = 0.771 (Tourists appreciate the country's cultures and lifestyle), while in F1.2 the variable that has the highest value is Q4.5 = 0.824 (Tourists support and preserve the cleanliness of the country's environment); both of these variables have great importance for the economic and financial development of Albania and Kosovo. According to the Friedman test rankings, all variables are important in the model, while the highest values are in growth and economic–financial development through tourism, as in the variable Q4.4 = 5.53, where according to the perceptions of residents, tourists focus on destinations and popular points of the country, while the variable that has the least value is Q4.5 = 4.04, tourists support the cleanliness of the environment of the country. In this case, it is recommended to increase the importance of this variable for both countries by creating a strategy for awareness of tourists in preserving the facilities in the tourist destinations where they would like to visit. According to the test KMO = 843, the data are very reliable and suitable for analysis with a value variance of 79.71%. According to ANOVA with Tukey's Test ($p = 0.000$) the result is statistically significant because there is a difference between the importance of variables in economic and financial development in both countries through tourism based on the effects of perceptions of local residents. Hotelling's T-Squared ($p = 0.000$) emphasizes the importance of variables in the model through their percentage. According to Cronbach's Alpha (0.839 apo ≈ 84%) it is emphasized that the data are very reliable and all variables should be included in the model regarding the impact of tourism on economic and financial development. The Guttman Split-Half Coefficient (0.824 apo ≈ 82%) and Split-half model Spearman–Brown Coefficient (0.827 apo ≈ 83%), further prove the importance of the variables in the model. According to Guttman Model Lambda (0.859; 0.867; 0.839; 0.824; 0.852; 0.879), all coefficients have high values and the meter is very reliable (apo ≈ 85%). According to the Parallel model, both coefficients again have high reliability (0.839; 0.846; apo ≈ 85%). According to the Strict Parallel model, the reliability of the coefficients is too high (0.803; 0.815; apo ≈ 80%). It is emphasized that the perceptions of residents about the behavior of tourists affect the economic and financial development through tourism in both countries.

Table 3 presents the results for the importance of the five variables from the Rotated Component Matrix to the factor analysis for subfactor F1.1.(0.779; 0.727; 0.691; 0.490; 0.434) as well as results from the Friedman test rankings (6.00; 5.95; 4.94; 4.89; 4.03). According to matrices of the factor analysis for subfactor F1.1, it is emphasized that the variables which have the greatest effect are F1.1 = 0.779; 0.727; 0.691; 0.590; 0.534. In. F1.1, the highest value is the variable Q4.2 = 0.771 (Tourists value the country's cultures and lifestyle); this variable is of great importance in economic and financial development. According to the Friedman ranking and test, all variables are important in the model, while the highest value is found in increasing economic and financial development through tourism. The highest rankingvariableis Q4.2 = 6.00, where according to residents' perceptions tourists appreciate the cultures and lifestyle of the country, while the variable with the lowest value is Q4.6 = 4.03, tourists do not face language barriers of the country; in this case, it is recommended to increase the importance of this variable for both countries by creating a strategy for greater knowledge of foreign languages for residents of both countries, facilitating communication for tourists. during purchases or other requests. According to the test KMO = 0.816, the data are very reliable and suitable for analysis, with a value variance of 80.81%. According to ANOVA with Tukey's Test ($p = 0.000$) the result is statistically significant because there is a difference between the importance of variables in the economic and financial development of the two countries through tourism based on the effects of perceptions of local residents. Hotelling's T-Squared ($p = 0.000$) emphasizes the importance of variables in the model through their percentage. According to Cronbach's Alpha (0.815 apo ≈ 82%), it is emphasized that the data are very reliable and all variables should be included in the model regarding the impact of tourism on economic and financial

development. The Guttman split-half Coefficient (0.824 apo ≈ 82%) and Split-half model Spearman–Brown Coefficient (0.827 apo ≈ 83%) further prove the importance of variables in the model. According to the Guttman Model Lambda (0.859; 0.867; 0.839; 0.824; 0.852; 0.879), all coefficients are of high value and the meter is very reliable (≈85%). According to the Parallel model, both coefficients have high reliability (0.839; 0.846) or ≈85%. According to the Strict Parallel model, the reliability of the coefficients is too high (0.803; 0.815, or ≈80%). It is emphasized that the perceptions of residents of the evaluation, adaptation, and visits of tourists affected the economic and financial development of tourism in both countries.

Table 3 presents the multiple regression for the first subfactor of F1, which explains 99% ($R^2$ = 0.989, Sig. = 0.000). This sub-factor depends on the independent variables (Q4.1; Q4.2; Q4.3; Q4.4; Q4.6), while 1% depends on other variables outside this model by random error. Adjusted R Square at a value of 0.985 indicates that 99% of the variables are related to the model, while according to the D-W test (1.119) the model is significant and the correlation is negative, which means that the standard error of the coefficient b or F1.1 is very small.

$$\hat{y} = \alpha_0 + \beta_1(Q4.2) + \beta_2(Q4.1) + \beta_3(Q4.3) + \beta_4(Q4.4) + \beta_5(Q4.6)$$
$$= 0.125 + 0.415x_1 + 0.356x_2 + 0.325x_3 + 0.121x_4 + 0.325x_5 + 0.01\mu$$

The *p* value is less than the 5% significance level, $H_0$ is rejected and $(\beta_1, \beta_2, \beta_3, \beta_4, \beta_5) \neq 0$.

Table 4 presents the results for the importance of the three variables from the Rotated Component Matrix to factor analysis for subfactor F1.2. (0.835; 0.754; 0.705) as well as the results from the Friedman test rankings (5.00; 6.05; 5.00). According to the matrix of the factorial analysis for subfactor F1.2, it is emphasized that the variables which have the greatest effect are F1.1 = 0.835; 0.754; 0.705. In F1.2, the highest value is the variable Q4.5 = 0.835 (Tourists support and preserve the cleanliness of the country's environment); this variable has great importance for economic and financial development in Albania and Kosovo. According to the Friedman ranking, all tested variables in the model are important, however, the highest value in growth and economic–financial development through tourism was found for the variable Q4.7 = 6005, where according to residents' perceptions tourists respect the rules and regulations of the country, while other variables had the same or smaller values (Q4.5 and Q4.8 = 5.00, tourists support and preserve the cleanliness of the country's environment and fulfill the country's requirements regarding COVID-19). In this case it is recommended to increase the importance of these variables for both countries by creating a tourist awareness strategy for environmental protection and compliance with COVID-19 rules. According to KMO = 0.714, the data are very reliable and suitable for analysis with a variance in the value of 68.76%. According to ANOVA with Tukey's Test (*p* = 0.000), the results are statistically significant, as there is a difference between the importance of variables in economic and financial development in both countries through tourism based on the effects of perceptions of local residents. Hotelling's T-Squared (*p* = 0.000) emphasizes the importance of variables in the model through their percentage. According to Cronbach's Alpha (0.745 or ≈75%), it is emphasized that the data are reliable and all variables should be included in the model regarding the impact of tourism on economic and financial development. The Guttman split-half Coefficient (0.726 or ≈73%) and Split-half model Spearman–Brown Coefficient (0.772 or ≈77%) further prove the importance of variables in the model. According to Guttman Model Lambda (0.730; 0.852; 0.845; 0.726; 0.871; 0.765), all coefficients have high values and the meter is very reliable, ≈80%. According to the Parallel model, both coefficients again have high reliability (0.745 and 0.752, or ≈75%). According to the Strict Parallel model, the reliability of the coefficients is too high (0.740 and 0.751, or ≈75%). It is emphasized that the observance of tourist rules by tourists affects the economic and financial development of both countries.

**Table 2.** Perceptions of AL and KO residents on tourist behavior (economic–financial analysis).

| | | | | |
|---|---|---|---|---|
| **FACTOR I** | | | | |
| **Perceptions of AL and KO Residents on Tourist Behavior (Economic–Financial Analysis)** | | | | |

| | | **Rotated Component Matrix** | | **Friedman Test** |
|---|---|---|---|---|
| | | **Component** | | |
| **Nr.** | **Variables** | **Sub. F1** | **Sub. F2** | **Ranks** |
| **Q4.2** | Tourists appreciate the cultures and lifestyle of the country | **0.771** | 0.081 | 5.04 |
| **Q4.1** | Tourists are friendly and respectful to the locals | **0.684** | 0.177 | 4.95 |
| **Q4.3** | Tourists are well adapted to the environment and climate of the country | **0.640** | 0.267 | 4.24 |
| **Q4.6** | Tourists do not face the country's language barriers | **0.514** | −0.283 | 3.89 |
| **Q4.4** | Tourists focus on popular destinations and points of the country | 0.495 | 0.003 | **5.53** |
| **Q4.5** | Tourists support and preserve the cleanliness of the country's environment | −0.045 | **0.824** | **4.04** |
| **Q4.8** | Tourists fulfill the country requirements regarding COVID-19 | 0.032 | **0.745** | 4.14 |
| **Q4.7** | Tourists respect the rules and regulations of the country | 0.387 | **0.648** | 4.16 |

| KMO Sig. | Total Variance | ANOVA with Tukey's Test for No additivity | Hotelling's T-Squared | Friedman Test | Cronbach's Alpha | Guttman Split-Half Coefficient | Split-half model Spearman–Brown Coefficient | Guttman Model Lambda | Parallel Model Reliability of Scale | Strict Parallel Model Reliability of Scale |
|---|---|---|---|---|---|---|---|---|---|---|
| | | | | | | | | | Reliability of Scale (Unbiased) | Reliability of Scale (Unbiased) |
| | | | | | | | | 0.859 | | |
| 0.843 | | Sig. | Sig. | Sig. | | | | 0.867 | 0.839 | 0.803 |
| | 79.71 | | | | 0.839 | 0.824 | 0.827 | 0.839 | | |
| | | | | | | | | 0.824 | | |
| 0.000 | | 0.000 | 0.000 | 0.000 | | | | 0.852 | 0.846 | 0.815 |
| | | | | | | | | 0.879 | | |

Table 3. (a) Tourists adjust, evaluate, and visit AL and KO (economic–financial analysis), (b) Multiple regression for F1.1 (Tourists adapt, evaluate, and visit AL and KO).

| (a) |
|---|
| **The First Sub-Factor of Factor 1** <br> **Perceptions of AL and KO Residents about the Behavior of Tourists (Economic–Financial Analysis)** |
| **Rotated Component Matrix** |

| Nr. | Variables | Tourists Adapt, Evaluate and Visit AL and KO | Fried Man Test |
|---|---|---|---|
| | | **Sub. F1** | **Ranks** |
| **Q4.2** | Tourists appreciate the country's cultures and lifestyle | **0.779** | **6.00** |
| **Q4.1** | The tourists are friendly and respectful to the locals | 0.727 | 5.95 |
| **Q4.3** | Tourists are well adapted to the environment and climate of the country | 0.691 | 4.94 |
| **Q4.4** | Tourists focus on popular destinations and points of the country | 0.590 | 4.89 |
| **Q4.6** | Tourists do not face the country's language barriers | 0.534 | **4.03** |

| KMO Sig. | Total Variance | ANOVA with Tukey's Test for No additivity | Hotelling's T-Squared | Friedman Test | Cronbach's Alpha | Guttman Split-Half Coefficient | Split-half model Spearman–Brown Coefficient | Guttman Model Lambda | Parallel Model – Reliability of Scale (Unbiased) | Strict Parallel Model – Reliability of Scale (Unbiased) |
|---|---|---|---|---|---|---|---|---|---|---|
| **0.816** | | Sig. | Sig. | Sig. | | | | 0.892 | 0.865 | 0.815 |
| | 80.81 | | | | 0.815 | 0.877 | 0.815 | 0.829 | | |
| | | | | | | | | 0.815 | | |
| **0.000** | | 0.000 | 0.000 | 0.000 | | | | 0.877 | 0.878 | 0.824 |
| | | | | | | | | 0.821 | | |
| | | | | | | | | 0.899 | | |

| (b) |
|---|
| **Multiple Regression Analysis** |

| Model Summary | | | | | | Change Statistics-ANOVA | | | | |
|---|---|---|---|---|---|---|---|---|---|---|
| **Model** | **R** | **R$^2$** | **Adj. R$^2$** | **Std. Error.** | **R Sq.** | **F** | **Df. 1** | **Df.2** | **Sig.** | **Durbin-Watson** |
| 1 | 0.982 | 0.989 | 0.985 | 0.0321 | 0.989 | 267.142 | 4 | 18 | 0.000 | 1.119 |

**Table 4.** (**a**) Tourists respect the tourist rules in AL and KO (economic–financial analysis); (**b**) Multiple regression for F1.2 (Tourists follow the tourist rules in AL and KO).

| (a) |
|---|

**The Second Sub-Factor of Factor 1**
**Perceptions of AL and KO Residents on Tourist Behavior (Economic-Financial Analysis)**

**Rotated Component Matrix**

| Nr. | Variables | Tourists Respect the Tourist Rules in AL and KO | Friedman Test |
|---|---|---|---|
| | | Sub. F2 | Ranks |
| **Q4.5** | Tourists support and preserve the cleanliness of the country's environment | **0.835** | **5.00** |
| **Q4.7** | Tourists respect the rules and regulations of the country | 0.754 | **6.05** |
| **Q4.8** | Tourists fulfill the country requirements regarding COVID-19 | **0.705** | **5.00** |

| KMO Sig. | Total Variance | ANOVA with Tukey's Test for No additivity | Hotelling's T-Squared | Friedman Test | Cronbach's Alpha | Guttman Split-Half Coefficient | Split-half model Spearman–Brown Coefficient | Guttman Model Lambda | Parallel Model Reliability of Scale / Reliability of Scale (Unbiased) | Strict Parallel Model Reliability of Scale / Reliability of Scale (Unbiased) |
|---|---|---|---|---|---|---|---|---|---|---|
| 0.714 | | Sig. | Sig. | Sig. | | | | 0.730 | | 0.740 |
| | | | | | | | | 0.852 | 0.745 | 0.751 |
| | 68.76 | | | | 0.745 | 0.726 | 0.772 | 0.845 | | |
| | | | | | | | | 0.726 | | |
| 0.000 | | 0.007 | 0.000 | 0.000 | | | | 0.871 | 0.752 | 0.752 |
| | | | | | | | | 0.765 | | |

| (b) |
|---|

**Multiple Regression Analysis**

| Model Summary | | | | | Change Statistics-ANOVA | | | | | |
|---|---|---|---|---|---|---|---|---|---|---|
| **Model** | **R** | **R²** | **Adj. R²** | **Std. Error.** | **R Sq.** | **F** | **Df. 1** | **Df.2** | **Sig.** | **Durbin-Watson** |
| 1 | 0.882 | 0.887 | 0.872 | 0.0811 | 0.887 | 674.041 | 4 | 27 | 0.000 | 1.623 |

Table 4 presents multiple regression for the second subfactor of F1, where it explains that 89% ($R^2$ = 0.887, Sig. = 0.000) of this sub-factor depends on the independent variables (Q4.5; Q4.7; Q4.3; Q4.8), while 11% depends on other variables outside of this model through random error. The Adjusted R Square value of 0.872 shows that 87% of the variables are related to the model, while according to the D-W test (1.623) the model is significant and correlation is negative, which means that the standard error of the coefficient b or F1.2 is very small.

$$\hat{y} = \alpha_0 + \beta_1(Q4.5) + \beta_2(Q4.8) + \beta_3(Q4.7) = 0.179 + 0.311x_1 + 0.316x_2 + 0.213x_3 + 0.11\mu$$

The $p$-value is less than the 5% significance level, thus, H$_0$ is rejected and $(\beta_1, \beta_2, \beta_3) \neq 0$.

### 4.2. Results for Factor II: Perceptions of Residents of AL and KO on the Role of Tourism (Economic–Financial Analysis)

Factor II (F2) or the effects of perceptions in Albania and Kosovo on the role of tourism (Figure 4) (impact of analysis and opportunities for economic and financial development) include three sub-factors, as follows:

- F2.1 = Possibilities of economic–financial development through tourism in AL and KO
- F2.2 = Economic–financial improvement through tourism in AL and KO
- F2.3 = Growth and well-being of households through tourism in AL and KO

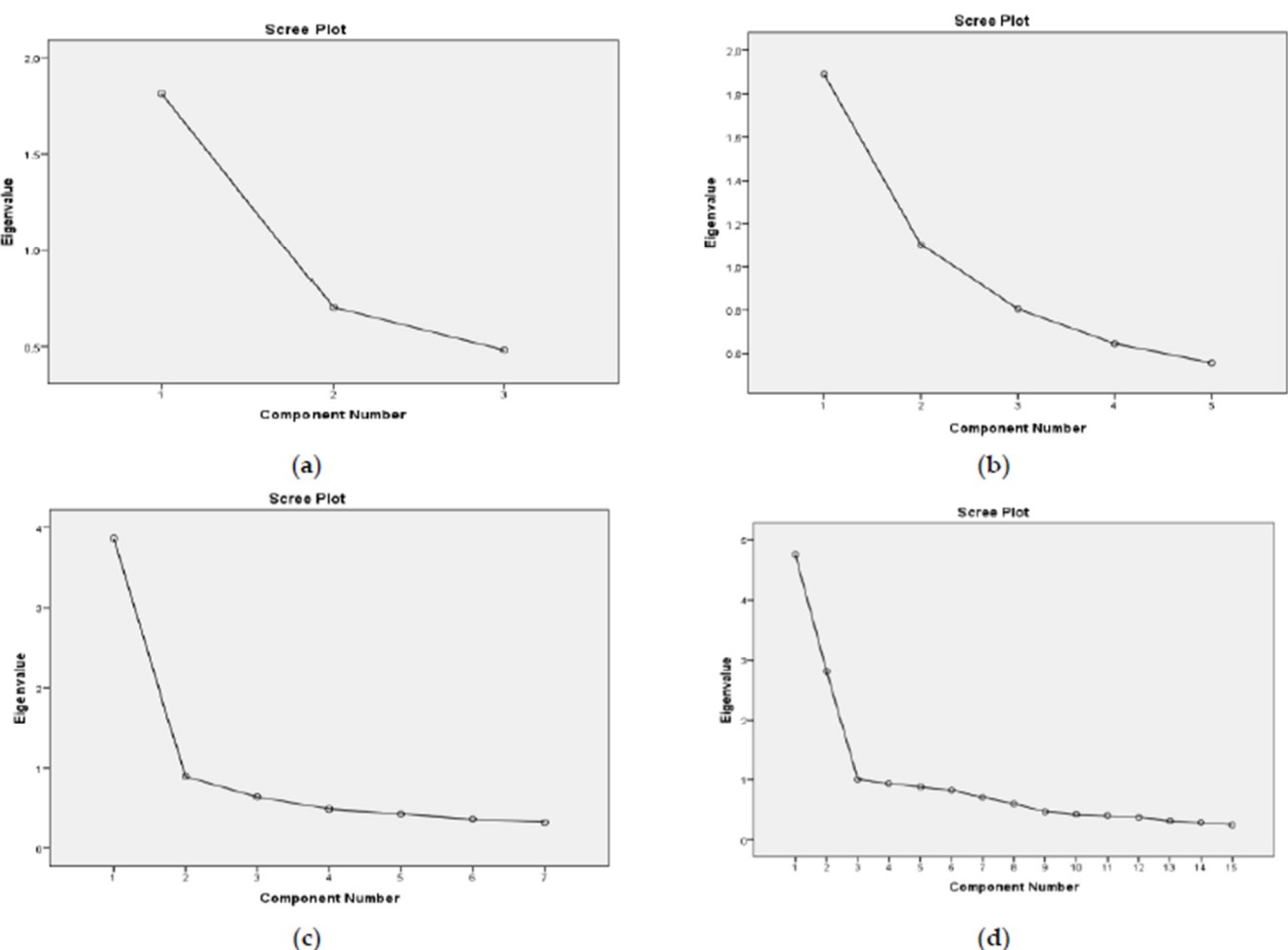

**Figure 4.** (**a**–**d**) Four Factor II (sub-factors 1, 2, 3, and 4), economic–financial analysis in AL and KO through tourism perceptions to the impact of tourism.

Table 5 presents the results for the importance of fifteen variables from Rotated Component Matrix to factor analysis for F2 and subfactors F2.1., F2.2 and F2.3 (0.825; 0.785; 0.763; 0.744; 0.636; 0.588; 0.563; 0.293; 0.091; 0.274; 0.221; 0.415; 0.021; 0.163; −0.245; 0.124; 0.118; 0.074; 0.148; 0.086; 0.482; 0.261; 0.712; 0.677; 0.665; 0.530; 0.451; 0.071; 0.249; 0.309; 0.021; −0.189; 0.210; −0.113; 0.069; −0.238; −0.413; 0.169; 0.359; 0.006; 0.384; 0.130; 0.791; 0.765; 0.567) as well as the result from the Friedman test rankings (9.20; 9.82; 9.80; 8.99; 9.33; 8.58; 9.72; 7.71; 7.22; 6.76; 5.37; 9.10; 6.29; 8.41; 3.69). According to the matrix in the factorial analysis for all three subfactors (F2.1, F2.2, and F2.3) it is emphasized that the variables which have effect are (F1.1 = 0.825; 0.785; 0.763; 0.744; 0.636; 0.588; 0.563; 0.293; 0.091; 0.274; 0.221; 0.415; 0.021; 0.163; −0.245), (F2.2 = 0.124; 0.118; 0.074; 0.148; 0.086; 0.482; 0.261; 0.712; 0.677; 0.665; 0.530; 0.451; 0.071; 0.249; 0.309), and (F2.3 = 0.021; −0.189; 0.210; −0.113; 0.069; −0.238; −0.413; 0.169; 0.359; 0.006; 0.384; 0.130; 0.791; 0.765; 0.567). In F2.1, the highest value is the variable Q5.5 =, 825 (Supports local shops, services, festivals/events), while the highest value in F2.2 is the variable Q5.8 = 0.712 (Creates overcrowding on roads/shops/vehicles of transport) and in F2.3 the highest value is the variable Q5.12 = 0.791 (Employment of residents of localities). All three variables are of great importance to the economic and financial development of Albania and Kosovo. According to the Friedman ranking and test, all variables are important in the model, while the highest value in terms of growth and economic–financial development through tourism was for the variable Q5.3 = 9.82, where according to redidents' perceptions tourism affects the attraction of investments in attractive areas, while the variable that has the lowest value is Q5.15 = 3.69, tourism requires security and development. In this case, it is recommended that the importance of this variable be increased for both countries by creating a strategy for attracting new investors and creating conditions and security for both investors and tourists by the governing bodies. According to KMO = 0.818 test, the data are very reliable and suitable for analysis with a value variance of 77.16%. According to ANOVA with Tukey's Test ($p = 0.000$), the results are statistically significant because there is a difference between the importance of variables in economic and financial development of the two countries through tourism based on the effects of perceptions of local residents regarding the role of tourism. Hotelling's T-Squared ($p = 0.000$) emphasizes the importance of variables in the model through their percentage. According to Cronbach's Alpha (0.848 or ≈ 85%), it is emphasized that the data are very reliable and all variables should participate in the model regarding the role of tourism in development and economic–financial opportunities. The Guttman split-half Coefficient (0.810 or ≈81%) and Split-half model Spearman–Brown Coefficient (0.812 or ≈81%) further prove the importance of the variables in the model. According to Guttman Model Lambda (0.753; 0.826; 0.806; 0.710; 0.799; 0.865), all coefficients have high values and the meter is very reliable (≈80%). According to Parallel model, both coefficients again have high reliability (0.861; 0.888, or ≈87%). According to the Strict Parallel model, the reliability of the coefficients is too high (0.811; 0.797, or ≈80%). It is emphasized that the perceptions of residents about the role of tourism affect the development and economic–financial opportunities in both countries.

**Table 5.** Perceptions of residents of AL and KO on the role of tourism (economic–financial analysis).

| | | Factor II<br>Perceptions of Residents of AL and KO on the Role of Tourism (Economic–Financial Analysis) | | | |
|---|---|---|---|---|---|
| | | Rotated Component Matrix | | | Friedman Test |
| Nr. | Variables | Component | | | |
| | | Sub. 1 | Sub. 2 | Sub. 3 | Ranks |
| Q5.5 | Supports local stores, services, festivals/events | **0.825** | 0.124 | 0.021 | 90.20 |
| Q5.3 | Attract investments in attractive areas | **0.785** | 0.118 | −0.189 | **90.82** |
| Q5.6 | Creates intercultural relations | **0.763** | 0.074 | 0.210 | 90.80 |
| Q5.2 | Creates more jobs for natives | **0.744** | 0.148 | −0.113 | 80.99 |
| Q5.7 | Supports the protection of cultural features and heritage | **0.636** | 0.086 | 0.069 | 90.33 |
| Q5.4 | Creates a market for local products | **0.588** | 0.482 | −0.238 | 80.58 |
| Q5.1 | Provides an economic contribution and financial stability | **0.563** | 0.261 | −0.413 | 90.72 |
| Q5.8 | Creates overcrowding on roads/shops/vehicles | 0.293 | **0.712** | 0.169 | 70.71 |
| Q5.9 | Income growth | 0.091 | **0.677** | 0.359 | 70.22 |
| Q5.11 | Improve transport availability | 0.274 | **0.665** | 0.006 | 60.76 |
| Q5.14 | Improve the livelihood of the locals | −0.221 | **0.530** | 0.384 | 50.37 |
| Q5.13 | Contribute to a lively and positive atmosphere | 0.415 | **0.451** | 0.130 | 90.10 |
| Q5.12 | Employment of local residents | 0.021 | 0.071 | **0.791** | 60.29 |
| Q5.10 | Increasing profits from the sale of goods and performance of services | 0.163 | 0.249 | **0.765** | 80.41 |
| Q5.15 | Security and development | −0.245 | 0.309 | **0.567** | **30.69** |

| KMO Sig. | Total Variance | ANOVA with Tukey's Test for No additivity | Hotelling's T-Squared | Friedman Test | Cronbach's Alpha | Guttman Split-Half Coefficient | Split-half model Spearman–Brown Coefficient | GuttmanModel Lambda | Parallel Model<br>Reliability of Scale (Unbiased) | Strict Parallel Model<br>Reliability of Scale (Unbiased) |
|---|---|---|---|---|---|---|---|---|---|---|
| 0.818 | | Sig. | Sig. | Sig. | | | | 0.753 | 0.861 | 0.811 |
| | | | | | | | | 0.826 | | |
| | | | | | | | | 0.806 | | |
| | 77.16 | | | | 0.848 | 0.810 | 0.812 | 0.710 | | |
| 0.000 | | 0.000 | 0.000 | 0.000 | | | | 0.799 | 0.888 | 0.797 |
| | | | | | | | | 0.865 | | |

Table 6 presents the results for the importance of seven variables from the Rotated Component Matrix to factor analysis for subfactor F2.1. (Q5.5; Q5.3; Q5.2; Q5.4; Q5.6; Q5.1; Q5.7) as well as results from the Friedman test rankings (3.85; 4.29; 3.78; 3.58; 4.23; 4.25; 4.02). According to the matrix in the factor analysis for subfactor F2.1, it is emphasized that the variables that have an effect are F2.1 = 0.814; 0.810; 0.787; 0.753; 0.721; 0.677. In F2.1, the highest value is the variable Q5.5 = 0.814 (Supports local stores, services, festivals/events); this variable is of great importance in economic–financial development through tourism. According to the Friedman ranking and test, all variables are important in the model, with the highest value in growth and economic–financial development through tourism for the variable Q5.3 = 4.29, where according to the perceptions of the residents tourism attracts investments in attractive areas, while the variable that has the lowest value is Q5.4 = 3.58, creates a market for local products. In this case, it is recommended to increase the importance of this variable for both countries by creating a strategy to create a market to sell local products to tourists. According to KMO = 0.877 test, the data are very reliable and suitable for analysis with a value variance of 85.22%. According to ANOVA with Tukey's Test (*p* = 0.004), the results are statistically significant because there is a difference between the importance of variables in economic and financial development in both countries through tourism based on the effects of perceptions of local residents. Hotelling's T-Squared (*p* = 0.000) emphasizes the importance of variables in the model through their percentage. According to Cronbach's Alpha (0.858 or ≈86%), it is emphasized that the data are very reliable and all variables should be included in the model regarding the impact of tourism on economic and financial development. According to Guttman split-half Coefficient (0.851 or ≈85%) and Split-half model Spearman–Brown Coefficient (0.889 or ≈89%) tests, of variables in the model are important. According to Guttman Model Lambda (0.736; 0.862; 0.858; 0.751; 0.841; 0.855), all coefficients have high values and the meter is very reliable (≈85%). According to the Parallel model, both coefficients again have high reliability (0.858; 0.861, or ≈86%). According to the Strict Parallel model, the reliability of the coefficients is too high (0.854; 0.858, or ≈85%). It is emphasized that the perceptions of the residents for the possibilities of economic and financial development through tourism in Albania and Kosovo are very important.

Table 6 presents the multiple regression for the first subfactor of F2.1, where it explains that 98% ($R^2$ = 0.981, Sig. = 0.000) of this sub-factor depends on the independent variables (Q5.5; Q5.3; Q5.2; Q5.4; Q5.6; Q5.1; Q5.7), while 2% depends on other variables outside this model by random error. An Adjusted R Square with a value of 976 indicates that 98% of the variables are related to the model, while according to the D-W (1.420) test the model is significant and the correlation is negative, which means that the standard error of coefficient b or F2.1 is too small.

$$\hat{y} = \alpha_0 + \beta_1(Q5.5) + \beta_2(Q5.3) + \beta_3(Q5.2) + \beta_4(Q5.4) + \beta_5(Q5.6) + \beta_6(Q5.1) + \beta_7(Q5.7)$$
$$= 0.125 + 0.391x_1 + 0.241x_2 + 0.174x_3 + 0.186x_4 + 0.276x_5 + 0.02\mu$$

The *p* value is less than the 5% significance level, $H_0$ is rejected, and $(\beta_1, \beta_2, \beta_3, \beta_4, \beta_5, \beta_6, \beta_7) \neq 0$.

Table 7 presents the results for the importance of the four variables from the Rotated Component Matrix to the factor analysis for subfactor F2.2 (Q5.11; Q5.13; Q5.9; Q5.14) as well as the results from the Friedman test rankings (2.38; 3.10; 2.54; 1.98). According to the matrix in the factor analysis for subfactor F2.2, it is emphasized that the variables that have an effect are F2.2 = 0.777; 0.722; 0.716; 0.555. In F2.2, the highest value is the variable Q5.11 = 0.777 (Improves the availability of transport); this variable is of great importance in economic and financial development. According to the Friedman rankings and test, all variables are important in the model, while the highest value in growth and economic–financial development through tourism is for the variable Q5.13 = 3.10, where according to the perceptions of the residents tourism contributes to a lively and positive atmosphere, while the variable with the smallest value is Q5.14 = 1.98, improves the

livelihood of the locals. In this case, it is recommended the importance of this variable for both countries be increased by creating a strategy to increase economic well-being through tourism. According to KMO = 755 test, the data are reliable and suitable for analysis with a value variance of 59.84%. According to ANOVA with Tukey's Test ($p = 0.000$) the results are statistically significant, because there is a difference between the importance of variables in economic and financial development in both countries through tourism based on the effects of perceptions of local residents. Hotelling's T-Squared ($p = 0.000$) emphasizes the importance of variables in the model through their percentage. According to Cronbach's Alpha (0.812 or ≈81%), it is emphasized that the data are very reliable and all variables should be included in the model regarding the impact of tourism on economic and financial development. The Guttman Split-Half Coefficient (0.704 or ≈74%) and Split-half model Spearman–Brown Coefficient (0.776 or ≈76%) prove the importance of variables in the model. According to Guttman Model Lambda (0.770; 0.721; 0.712; 0.804; 0.819; 0.884), all coefficients have high values and the meter is very reliable (≈80%). According to the Parallel model, both coefficients again have high reliability (0.712; 0.718, or ≈70%). According to the Strict Parallel model, the reliability of the coefficients is too high (0.763; 0.773, or ≈75%). It is emphasized that the economic and financial improvement through tourism in Albania and Kosovo is quite important.

Table 7 presents multiple regression for the second subfactor of F2.2 where it explains that 99% ($R^2 = 0.992$, Sig. = 0.000) this sub-factor depends on the independent variables (Q5.11; Q5.13; Q5.9; Q5.14), while 1% depends on other variables outside this model by random error. Adjusted R Sq. at a value of 985 indicates that 99% of the variables are related to the model, while according to test D-W (1.171) the model is significant and the correlation is negative, which means that the standard error of the coefficient b or F2.2 is very small.

$$\hat{y} = \alpha_0 + \beta_1(Q5.11) + \beta_2(Q5.13) + \beta_3(Q5.9) + \beta_4(Q5.14)$$
$$= 0.117 + 0.421x_1 + 0.178x_2 + 0.236x_3 + 0.1157x_4 + 0.01\mu.$$

The $p$ value is less than the 5% significant level, $H_0$ is rejected, and $(\beta_1, \beta_2, \beta_3, \beta_4,) \neq 0$.

Table 8 presents the results for the importance of the three variables from the Rotated Component Matrix to factor analysis for subfactor F2.3. (0.828; 0.800; 0.700) as well as the results from the Friedman test rankings (2.05; 2.52; 1.43). According to the matrix of the factorial analysis for subfactor F2.3, it is emphasized that the variables which have effect are F2.3 = 0.828; 0.800; 0.700. In F2.3, the highest value is the variable Q5.12 = 0.828 (Employment of residents of localities); this variable is of great importance in economic and financial development. According to the Friedman ranking and test, all variables are important in the model, while the highest value in growth and economic–financial development through tourism is for the variable Q5.10 = 2.52, where according to the perceptions of the residents if tourism is developed there will be an increase in profits from the sale of goods and the provision of services, while the variable with the lowest value is Q5.15 = 1.43, security and development; in this case, it is recommended to increase the importance of this variable for both countries through the creation of a strategy for security and development of countries through tourism. According to KMO = 0.777 test, the data are reliable and suitable for analysis with a value variance of 60.48%. According to ANOVA with Tukey's Test ($p = 0.000$), the results are statistically significant because there is a difference between the importance of variables in economic and financial development in both countries through tourism based on the effects of perceptions of local residents. Hotelling's T-Squared ($p = 0.000$) emphasizes the importance of variables in the model through their percentage. According to Cronbach's Alpha (0.818 or ≈82%), it is emphasized that the data are very reliable and all variables should be included in the model regarding the impact of tourism on economic and financial development. The Guttman Split-Half Coefficient (0.824 or ≈82%) and Split-half model Spearman–Brown Coefficient (0.876 or ≈88%) prove the importance of variables in the model. According to Guttman Model Lambda (0.745; 0.872; 0.867; 0.724; 0.784; 0.686; 0.879), all coefficients have high values and

the meter is very reliable ($\approx$75%). According to the Parallel model, both coefficients again have high reliability (0.867; 0.873, or $\approx$86%). According to the Strict Parallel model, the reliability of the coefficients is too high (0.828; 0.845, or $\approx$80%). It is emphasized that if the tourism of both countries is developed, there will be growth and well-being in households.

Table 8 Shows multiple regression for the third subfactor of F2.3 where it explains that 88% ($R^2$ = 0.879, Sig. = 0.000) of this sub-factor depends on the independent variables (Q5.12; Q5.10; Q5.15), while 12% depends on other variables outside this model by random error. The Adjusted R Square value of 0.878 shows that 88% of the variables are related to the model, while according to the D-W test (1.318) the model is significant and the correlation is negative, which means that the standard error of coefficient b or F2.3 is very small.

$$\hat{y} = \alpha_0 + \beta_1(Q5.12) + \beta_2(Q5.10) + \beta_3(Q5.15)$$
$$= 0.149 + 0.347x_1 + 0.317x_2 + 0.299x_3 + 0.12\mu$$

The *p* value is less than the 5% significance level, $H_0$ is rejected, and $(\beta_1, \beta_2, \beta_3) \neq 0$.

According to Table 9, in both factors F1 and F2 as well as sub-factors of both factors (F1.1 and F1.2) and (F2.1, F2.2, and F2.3) emphasize the validation of the alternative hypothesis emphasizing the effect of respondents' perceptions related to tourism and its impact on economic and financial development in both countries. In this case, tourism is of great importance and is a very important and necessary opportunity for the economic and financial development of Albania and Kosovo. There is a difference between the variables according to the perceptions of the respondents, which means that certain variables have a greater weight in the economic and financial development of the countries through tourism, whether maritime or mountain.

**Table 6.** (**a**) Possibilities of economic–financial development through tourism in AL and KO. (**b**) Multiple regression for F2.1 (Possibilities of economic-financial development through tourism in AL and KO).

| | | | (a) | |
|---|---|---|---|---|
| | | **The First Sub-Factor of Factor II** **Perceptions of Residents of AL and KO on the Role of Tourism (Economic-Financial Analysis)** | | |

**Rotated Component Matrix**

| Nr. | Variables | Possibilities of Economic-Financial Development through tourism in AL and KO | Friedman Test |
|---|---|---|---|
| | | Sub. F1 | Ranks |
| **Q5.5** | Supports local stores, services, festivals/events | **0.814** | 3.85 |
| **Q5.3** | Attract investments in attractive areas | 0.810 | **4.29** |
| **Q5.2** | Creates more jobs for natives | 0.787 | 3.78 |
| **Q5.4** | Creates a market for local products | 0.753 | **3.58** |
| **Q5.6** | Creates intercultural relations | 0.721 | 4.23 |
| **Q5.1** | Provides an economic contribution and financial stability | 0.677 | 4.25 |
| **Q5.7** | Supports the protection of cultural features and heritage | **0.617** | 4.02 |

| KMO Sig. | Total Variance | ANOVA with Tukey's Test for No additive | Hotelling's T-Squared | Friedman Test | Cronbach's Alpha | Guttman split-half Coefficient | Split-half model Spearman–Brown Coefficient | Guttman Model Lambd | Parallel Model Reliability of Scale (Unbiased) | Strict Parallel Model Reliability of Scale (Unbiased) |
|---|---|---|---|---|---|---|---|---|---|---|
| 0.877 | | Sig. | Sig. | Sig. | | | | 0.736 | | |
| | | | | | | | | 0.862 | 0.858 | 0.854 |
| | 85.22 | | | | 0.858 | 0.851 | 0.889 | 0.858 | | |
| | | | | | | | | 0.751 | | |
| 0.000 | | 0.004 | 0.004 | 0.000 | | | | 0.841 | 0.861 | 0.858 |
| | | | | | | | | 0.855 | | |

| | | | (b) | |
|---|---|---|---|---|

**Multiple Regression Analysis**

| | Model | | | | | | | | | R | |
|---|---|---|---|---|---|---|---|---|---|---|---|
| Model | R | R$^2$ | Adj. R$^2$ | Std. Error. | R Sq. | F | Df. 1 | Df.2 | Sig. | Durbin-Watson |
| 1 | 0.976 | 0.981 | 0.976 | 0.0531 | 0.981 | 1924.1 | 9 | 34 | 0.000 | 1.420 |

**Table 7.** (**a**) Economic–financial improvement through tourism in AL and KO; (**b**) Multiple regression for F2.2 Economic–financial improvement through tourism in AL and KO.

| (a) | | | | |
|---|---|---|---|---|
| **The Second Sub-Factor of Factor II** | | | | |
| **Perceptions of Residents of AL and KO on the Role of Tourism (Economic–Financial Analysis)** | | | | |

| | | **Rotated Component Matrix** | **Economic–Financial Improvement through Tourism in AL and KO** | **Friedman Test** |
|---|---|---|---|---|
| **Nr.** | | **Variables** | **Sub. F2** | **Ranks** |
| **Q5.11** | | Improves transport availability | **0.777** | 2.38 |
| **Q5.13** | | Contributes to a lively and positive atmosphere | 0.722 | **3.10** |
| **Q5.9** | | Income growth | 0.716 | 2.54 |
| **Q5.14** | | Improves the livelihood of the locals | **0.555** | **1.98** |

| KMO Sig. | Total Variance | ANOVA with Tukey's Test for No additivity | Hotelling's T-Squared | Friedman Test | Cronbach's Alpha | Guttman Split-Half Coefficient | Split-half model Spearman–Brown Coefficient | Guttman Model Lambda | Parallel Model Reliability of Scale / Reliability of Scale (Unbiased) | Strict Parallel Model Reliability of Scale / Reliability of Scale (Unbiased) |
|---|---|---|---|---|---|---|---|---|---|---|
| 0.755 | | Sig. | Sig. | Sig. | | | | 0.770 | 0.712 | 0.763 |
| | 59.84 | | | | 0.812 | 0.704 | 0.776 | 0.721 | | |
| | | | | | | | | 0.712 | | |
| | | | | | | | | 0.804 | | |
| 0.000 | | 0.000 | 0.000 | 0.000 | | | | 0.819 | 0.718 | 0.773 |
| | | | | | | | | 0.884 | | |

| (b) | | | | | | | | | | |
|---|---|---|---|---|---|---|---|---|---|---|
| **Multiple Regression Analysis** | | | | | | | | | | |
| **Model** | | | | | | **R** | | | | |
| **Model** | **R** | **R²** | **Adj. R²** | **Std. Error.** | **R Sq.** | **F** | **Df. 1** | **Df.2** | **Sig.** | **Durbin-Watson** |
| 1 | 0.981 | 0.992 | 0.985 | 0.0321 | 0.992 | 212,421 | 7 | 25 | 0.000 | 1.171 |

**Table 8.** (**a**) Growth and well-being of households through tourism in AL and KO. (**b**) Multiple regression for F2.3 Growth and well-being of households through tourism in AL and KO.

**(a)**

**Third Sub-Factor of Factor II**
**Perceptions of Residents of AL and KO on the Role of Tourism (Economic–Financial Analysis)**

**Rotated Component Matrix**

| Nr. | Variables | Growth and Well-Being of Households through tourism in AL and KO | Friedman Test |
|---|---|---|---|
| | | Sub. F3 | Ranks |
| **Q5.12** | Employment of residents of localities | **0.828** | 2.05 |
| **Q5.10** | Increasing profits from the sale of goods and the provision of services | 0.800 | **2.52** |
| **Q5.15** | Security and development | **0.700** | 1.43 |

| KMO Sig. | Total Variance | ANOVA with Tukey's Test for No additivity | Hotelling's T-Squared | Friedman Test | Cronbach's Alpha | Guttman Split-Half Coefficient | Split-half model Spearman–Brown Coefficient | Guttman Model Lambda | Parallel Model Reliability of Scale / Reliability of Scale (Unbiased) | Strict Parallel Model Reliability of Scale / Reliability of Scale (Unbiased) |
|---|---|---|---|---|---|---|---|---|---|---|
| 0.777 | | Sig. | Sig. | Sig. | | | | 0.745 | | |
| | | | | | | | | 0.872 | 0.867 | 0.828 |
| | 600.48 | | | | 0.818 | 0.824 | 0.8776 | 0.867 | | |
| | | | | | | | | 0.724 | | |
| 0.000 | | 0.000 | 0.000 | 0.000 | | | | 0.784 | 0.873 | 0.845 |
| | | | | | | | | 0.686 | | |

**(b)**

**Multiple Regression Analysis**

| | Model Summary | | | | | Change Statistics-ANOVA | | | | |
|---|---|---|---|---|---|---|---|---|---|---|
| Model | R | $R^2$ | Adj. $R^2$ | Std. Error. | R Sq. | F | Df. 1 | Df.2 | Sig. | Durbin-Watson |
| 1 | 0.864 | 0.879 | 0.878 | 0.0415 | 0.879 | 12,401 | 6 | 19 | 0.000 | 1.318 |

**Table 9.** Verification of hypotheses.

| Factors | Sub-Factors | Multiple regression Mathematical Equation Clarification of Hypotheses |
|---|---|---|
| FACTOR I Perceptions of AL and KO residents on tourist behavior (economic–financial analysis) | The first sub-factor of Factor I Perceptions of AL and KO residents about the behavior of tourists (economic-financial analysis) | $\hat{y} = \alpha_0 + \beta_1(Q4.2) + \beta_2(Q4.1) + \beta_3(Q4.3) + \beta_4(Q4.4) + \beta_5(Q4.6) = 0.125 + 0.415x_1 + 0.356x_2 + 0.325x_3 + 0.121x_4 + 0.325x_5 + 0.01\mu$ The $p$ value is less than the 5% significance level, $H_0$ is rejected and accepted $(\beta_1, \beta_2, \beta_3, \beta_4, \beta_5) \neq 0$. |
| | The second sub-factor of factor I Perceptions of AL and KO residents on tourist behavior (economic–financial analysis) | $\hat{y} = \alpha_0 + \beta_1(Q4.5) + \beta_2(Q4.8) + \beta_3(Q4.7) = 0.179 + 0.311x_1 + 0.316x_2 + 0.213x_3 + 0.11\mu$ The $p$ value is less than the 5% significance level, $H_0$ is rejected, and $(\beta_1, \beta_2, \beta_3) \neq 0$. |
| Factor II Perceptions of residents of AL and KO on the role of tourism (economic–financial analysis) | The first sub-factor of factor II Perceptions of residents of AL and KO on the role of tourism (economic–financial analysis) | $\hat{y} = \alpha_0 + \beta_1(Q5.5) + \beta_2(Q5.3) + \beta_3(Q5.2) + \beta_4(Q5.4) + \beta_5(Q5.6) + \beta_6(Q5.1) + \beta_7(Q5.7) = 0.125 + 0.391x_1 + 0.241x_2 + 0.174x_3 + 0.186x_4 + 0.276x_5 + 0.02\mu$ The $p$ value is less than the 5% s significant level, $H_0$ is rejected, and $(\beta_1, \beta_2, \beta_3, \beta_4, \beta_5, , \beta_6, \beta_7) \neq 0$. |
| | The second sub-factor of factor II Perceptions of residents of AL and KO on the role of tourism (economic–financial analysis) | $\hat{y} = \alpha_0 + \beta_1(Q5.11) + \beta_2(Q5.13) + \beta_3(Q5.9) + \beta_4(Q5.14) = 0.117 + 0.421x_1 + 0.178x_2 + 0.236x_3 + 0.1157x_4 + 0.01\mu.$ The $p$ value is less than the 5% significant level, $H_0$ is rejected, and $(\beta_1, \beta_2, \beta_3, \beta_4,) \neq 0$. |
| | Third sub-factor of factor II Perceptions of residents of AL and KO on the role of tourism (economic–financial analysis) | $\hat{y} = \alpha_0 + \beta_1(Q5.12) + \beta_2(Q5.10) + \beta_3(Q5.15) = 0.149 + 0.347x_1 + 0.317x_2 + 0.299x_3 + 0.12\mu$ The $p$ value is less than the 5% significance level, $H_0$ is rejected, and $(\beta_1, \beta_2, \beta_3) \neq 0$. |

## 5. Discussion

Today, the tourism industry plays an important role in the economic and financial development of countries and its impact is greater than ever. Globally, travel and tourism are significant contributors to a leading sector for job creation and socio-economic and cultural development worldwide [2]. Despite the various crises and adversities experienced, tourism is among the sectors that have been growing rapidly all over the world in recent years [50]. Here, essential variables in the tourism literature are brought together in one study and for the first time the effect of perception of tourists through tourism and the impact on the economic and financial development for the countries of Albania and Kosovo is presented. From now on, Albania and Kosovo should focus on promotion of an image of a quality destination, non-massified, and unpolluted, which are all factors highly appreciated by tourists [51]. This paper examines the tourists' perceptions that have greater demand for small regions, cities, and places where they can have unique experiences. Perceptions associated with shopping for local products and quality of service while on vacation are likely to impact on tourists' shopping behavior [52]. The demographic characteristics of tourists which may have impacts on effect of perceptions on tourism include individuals' age, gender, and prior travel experience [33].

The present study shows that special attention should be paid to the following factors: greater knowledge of foreign languages for residents of both countries, which can facilitate communication for tourists during purchases or other requests; and creating conditions and security for both investors and tourists by government bodies. According to the Friedman ranking and test, the variable that has the lowest value is Q4.6 = 4.03 (tourists do not face language barriers of the country); in this case, it is recommended to increase the importance of this variable for both countries by creating a strategy for greater knowledge of foreign languages for residents of both countries facilitating communication for tourists during purchases or other requests. Numerous econometric models can be adopted to explore factors leading to impacts and opportunities for economic and financial development in Albania and Kosovo. Moreover, the index and its sub-indices can serve as independent and control variables in various econometric models. Global challenges has had impacts on tourism and travel, that are a major contributor to the service industry worldwide [53] Today, many developing countries are in a wild competition in order to attract international visitors from developed countries; thus, they need to improve their business environment. As in many developing countries, the tourism industry in Albania and Kosovo is one of the leading sectors of the economy. Restarting tourism is very important for countries that are highly economically dependent on this industry. These countries should find an effective solution and use measures for protecting and supporting the community [54]. Our results confirm that respecting the country's cultures and lifestyle has an impact on the economic and financial development of these two countries. In a travel context, subjective perceptions can affect tourists' destination choices and travel behavior [38]. Our results highlight the importance for both countries of creating a strategy for promoting awareness in tourists about preserving the facilities in the tourist destinations where they would like to visit. Although this study focuses on Albania and Kosovo, the present research is interesting to the academia in general and the contributes to the literature regarding countries where the economy is interlinked with tourism.

## 6. Conclusions

Today, tourism plays an important role in the economic and financial development of countries, and its impact is greater than ever. Despite many problems, recent years have created an opportunity for the development of domestic tourism in Albania and Kosovo, if the host communities can remodel their tourism offering in order to comply with changing demands of visitors and refocus their offerings to accommodate new (maybe domestic) markets or to move out of tourism; certain destinations in these two countries will undoubtedly reconsider the nature of their tourism industry and focus more on local

and more sustainable forms of tourism, creating conditions to restart domestic economies and education systems. Regarding the econometric models which are used in our paper, the results show that the effect of perception on the behavior of tourists in these two countries has an impact in economic and financial development. Based on our findings, the perception of tourists regarding respecting the cultural of communities are of great importance in economic and financial development in Albania and Kosovo. It is argued that local residents should be encouraged to visit local attractions because of their awareness of the local COVID-19 pandemic situation, and thus might be more confident to engage in tourism-related activities with family in local areas.

In his research, we are based on the factors that have been taken in our study, and the hypotheses which are raised are confirmed. The results show that special attention should be paid to these factors: (a) awareness of tourists in preserving facilities in tourist destinations where they would like to visit; (b) greater knowledge of foreign languages for residents of both countries and facilitating communication for tourists during purchases or other requests; (c) attracting new investors and creating conditions and security for both investors and tourists by government bodies; (d) supporting the marketing and sale of local products for tourists; (e) the need for support from government bodies in terms of infrastructure in order to increase economic and financial well-being through tourism for both countries. Governments should provide more facilities and interventions to protect the environment, ensuring that tourists observe the rules, and adapt tourist destinations to ensure the viability and continuity of these destinations in the future. What should the role of governments be in the future of tourism and destinations' tourism policy-making and strategy? The results of this research reveal that the tourism sector is very sensitive and easily affected by global crises. For sustainable economic and financial growth and well-planned development, both public and private investments should be directed to areas of priority tourism development. In terms of sustainable development and regional competitiveness, Kosovo and Albania should follow development trends and be competitive. Even though this study focuses on Albania and Kosovo, the elements identified regarding the challenges and opportunities for other small-scale economies and countries are relevant to other places, such as other Balkan countries, who look to the tourism as a way to sustain their economic growth. Our findings reinforce the need for implementing strategies with a focus on sustainable development in tourism in both countries.

The need for assistance has increased in all tourism sectors. Interventions by state institutions have become necessary in order to alleviate the financial crisis as much as possible. Specifically, among other measures for tourism development, this study recommends that the governments of both countries should develop a significant strategy with specific proposals to improve tourism.

## 7. Recommendations

Based on all variables and our finding from all factors and sub-factors, we recommend:

a. Create a strategy to promote the awareness of tourists of the importance of preserving the facilities in the tourist destinations where they would like to visit.

b. The governments of Albania and Kosovo need to combine tourist attractions and activities into consolidated tourism products and promote them on the regional, European, and global markets. The promotion of the country as a safe and welcoming tourist destination with consolidated tourist offerings and authentic products is closely related to this element.

c. Create a strategy to promote greater knowledge of foreign languages for residents of both countries in order to facilitating communication for tourists. during purchases or other requests Both countries need to establishing a continuous professional training system for all human resources in the tourism industry.

d. Create a strategy for attracting new investors and creating conditions and security for both investors and tourists by the governing bodies.

e. Harmonization and standardization of other laws and sub-legal acts will help to to increase investment.

f. Create a strategy and support in marketing and selling local products to tourists. This promotional strategy should be developed with priority to digital marketing, online platforms, and apps which help communities to sell their local products.

g. Increase economic well-being through tourism in both countries.

h. Create a strategy for security and development in both countries through tourism.

i. It is strongly recommended for both countries that interested people have access to clear maps online to avoid tourists or even local residents having transportation problems.

This paper can help not only the state of Kosovo and the state of Albania, it can help all countries of the world to see the effects of perceptions on tourists and their impact on economic and financial development. As mentioned above, tourism has great importance for economic development all over the world, especially for those countries that are known for tourism.

The implications and limitations of this research are primarily that only a certain number of variables have been used, and only a few cities in the countries of Albania and Kosovo have been studied, as elaborated above. A major restriction was access to clear maps and tourist maps in English, according to the requirements of the reviewers. We had a lot of problems with this part of the maps. Therefore, other similar analyses could use a larger number of variables, more cities, or different countries, or even make comparisons. These possibilities can be elaborated in the future by authors from these or other countries around the world.

**Author Contributions:** Conceptualization, M.T. and E.L.; methodology, E.L. and M.T.; software, E.L. and M.T.; validation, E.L. and M.T.; formal analysis, E.L. and M.T.; investigation, E.L. and M.T.; resources, E.L. and M.T.; data curation, E.L. and M.T.; writing—original draft preparation, M.T. and E.L. writing—review and editing, M.T. and E.L.; visualization, M.T. and E.L.; supervision, E.L. and M.T. All authors have read and agreed to the published version of the manuscript.

**Funding:** This research received no external funding.

**Institutional Review Board Statement:** The study was conducted according to the guidelines of the Declaration of Helsinki and Ethical review and approval were waived for this study, for the reason that this study was conducted individually and independently by the institution where they work, respecting the anonymity of the interviewer.

**Informed Consent Statement:** Informed consent was obtained from all subjects involved in the study.

**Data Availability Statement:** The data used to support and prove the findings of this study are available from the corresponding author upon request.

**Conflicts of Interest:** The authors declare no conflict of interest.

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
