# Peer review of "The Effect of Perceptions on Tourism: An Econometric Analysis of the Impacts and Opportunities for Economic and Financial Development in Albania and Kosovo"

_sustainability, doi:10.3390/su14137659_

Round 1

Reviewer 1 Report

My most serious remark relates to the fact that in-depth literature review is missing. When trying to state the hypothesis it should be preceded by discussion already conducted by other authors. This, strongly advise to discuss contexts interesting for authors, and based on that to state the hypothesis. As a result I would like to see literature review concluded by hypotheses statement.

There is nothing about the methods. It looks like the authors decided to run SPSS with no in-depth understanding what is going on behind the software. I believe this is not true. However, a bit more detailed information about the methods is needed. And finally, apply only those of the methods which are significant from the perspective of testing the hypotheses stated.

Moreover, discussion on the questionnaire is missing as well. We could identify only the questions of the questionnaire presented in the Table 1. However, those of the questions are not justified by any discussion or literature review. Strongly advise to explain not only the issues of sampling but also merit part of the questionnaire. This should refer to literature review and hypotheses.

I appreciate that authors decided to propose some policy recommendations. However, it should be clearly presented as a separate section, not the conclusions. Please, read carefully requirements of Sustainability journal related to the section of conclusions.

As the paper is completely out of recent scientific discussion: no strong linkages with existing literature were evidenced; However, the topic seems to be interesting. If authors decide to start the research process once again but based on in-depth literature review, the paper could be rewritten and submitted once again.

Author Response

Dear reviewer,

We thank you endlessly for all your suggestions so that our paper is as valuable and quality as possible. We have tried in full to fulfill all the suggestions you have given. Your help was important to us. 

With appreciation and gratitude 

Authors.

Reviewer 2 Report

In the abstract, the number of interviews stated is 1002, however in the 2nd page it is mentioned 100. Please correct the one which is wrong of the two numbers;

The authors should not rely only on the quantitative methods based on the interviews, but should also support the study by qualitative methods to make the research more comprehensive and robust.

Introduction can be more elaborated by citing more examples such as the impact of Covid 19 on the main tourist cities, such as Paris, London and Istanbul...etc to make this research with an international impact;

Please specify if the interviews were done online or onsite.

The authors did not explain why they selected these 07 cities; (Tirana, Durres, Fier, Pristina, Peja, Gjakova, Istog). Whart are the criteria of selection with regard to tourism.

The authors need to put the reader in context. There is no introduction about the two countries Kosovo and Albania in relation to tourism. Not even a single map displaying the location of these two countries as well as the 07 selected cities;

In Setion 04 (Discussion) spelling mistake All trips and not all trip.

No pictures or illustrations supporting the argument, not even two pictures taken while undertaking the surveys in Kosovo and Albania;

The introduction and conclusion are too short. In the conclusion, the authors should focus on the merit of this research not only in Kosovo and Albania but globally.

Author Response

Dear reviewer,

We thank you endlessly for all your suggestions so that our paper is as valuable and quality as possible. We have tried in full to fulfill all the suggestions you have given. 

Your help was very important to us. 

With appreciation and gratitude 

Authors

Reviewer 3 Report

The subject of the paper is interesting and up-to-date.
However, a literature review that is missing from the text should be developed.
We have a short introduction describing the changes in tourism caused
by the pandemic and a section describing the research methodology right away.
In the part concerning the methodology, there is no information
about the scale used in the survey. There are no references to the works
of other authors.
The discussion is very poor. There are no comparisons of own research
findings with the research of other authors.
The conclusions should clearly emphasize the contribution of the article
to the development of the theory.
There is also no description of the limitations of the research carried out
and the directions for further research.

Author Response

Dear reviewer

We thank you endlessly for all your suggestions so that our paper is as valuable and quality as possible. We have tried in full to fulfill all the suggestions you have given.

Your help was very important to us. 

With appreciation and gratitude

Authors

Reviewer 4 Report

Dear authors,

I read your article, which addresses a topic of interest. In this context, major improvements are needed, such as:
-the summary is too long with too many details, I think you need to write it more concisely
- the novelty of your study must be added to the introduction
- the article does not have the "literature review" section, I recommend you add it.
- research hypotheses are not supported by the literature

In terms of methodology, improvements are needed such as:
-First of all, the arrangement in the table is inappropriate. The authors use .000 or, 000. The accepted notation is 0.000 or another number, respecting the same form with zero before the decimal point, followed by three decimals.
- The second point is misplaced commas and there are enough cases.
-Before arriving at the actual methodology, the authors do not present data of the studied population. Age, gender, nationality are optional, but may give the work a better look.
-Also, given that questionnaires were used applied in several cities, several countries, the authors could make a map of the distribution of the studied population. It's optional, but a big plus.
- It is not clear if the results obtained are the desired ones, the hypotheses are not discussed at all after obtaining the results. A table showing whether the hypotheses are being tested or not could help.
-The discussion part should also be used to discuss the results obtained and the implications of these results, instead discussing tourism in Albania and Kosovo. These should be discussed in the introduction, and the discussion section should be more concise in the end.
- argue the representativeness of the sample
- you can add the research tool in the attachments
-attention to the bibliography, there are quite a few references

These are just some of the goal setting shareware that you can use.

Good luck!

Author Response

(The authors gave the same response as above.)

Round 2

Reviewer 1 Report

I appreciate all efforts of authors to improve the quality of the paper. Still having some major remarks.

The quality of maps 1 and 2 should be improved. I suppose these are the copies from some online sources. I strongly advise the authors to elaborate own maps and cartographic analysis. Mind to use similar styles of cartographic presentation on both of the maps. It relates also to the issues of copyrights.

I do not know the reason that COVID-19 issues are major part of literature review. In fact, the pandemic is not the context of the analysis. The main problem you discuss is a perception of tourism as a tool for achieving developmental goals (environmental, social, and economic). Different contexts of that perception were mentioned. I strongly advise to rewrite the literature review. Do not put so much attention on COVID-19. Focus on contexts of perception you investigate: perception on tourists behaviour (how locals see tourists?), and perception of contribution of tourism to local development (how locals see tourism?). This is missing in your literature review. Moreover, there are no clear linkages between hypotheses and literature review. As I said before, literature review should be concluded by hypotheses statement. 

The section of results must be improved as well. You only present the results but do not interpret it. There is a lot of values and statistics but with no in-depth analysis and interpretation. As the result, the text is difficult to read and - to be honest - a bit boring.

Finally, when discussing policy recommendations, clear linkages between results and recommendations should be set. Thus, try to improve and elaborate the section of recommendations.

Author Response

Dear respected reviewer 

Thank you so very much for your help. We are very honored and grateful. 

With appreciation 

Authors 

Reviewer 2 Report

The authors have made major improvements to the paper.

Author Response

Dear respected reviewer 

Thank you very much. We are very honored and grateful 

With appreciation 

Authors 

Reviewer 3 Report

1.The aim of the paper should be clarified in the introduction 2. I have a problem with the evaluation of the literature review.
This section of the article should be located within a given theory
and provide an overview of publications related to that theory.
There are no references to the theory here.
There is brief information about the relationship between
the Covid pandemic and tourism. And then we immediately
have a description of the situation in Albania and Kosovo.

3.
The discussion should relate to the theory presented
in the part "Literature review". It should show what the research presented
in the paper confirms or what contribute to the existing theory.
This is missing, because in my opinion the part of "Literature Review"
does not relate to any theory.

4. Writing in my review that "there is no information about the scale used in the
survey" - I
meant the statements used in the LIkert scale in the questionnaire. .

5. Authors should clearly emphasize the contribution of the article
to the development of the theory. What was added in the "Conclusions"
section has the nature of practical implications, and not of a description
of the article's scientific contribution to an existing theory.

Author Response

Dear respected reviewer 

Thank you very much . We are very honored and grateful. 

With appreciation and gratitude 

Sincerely yours

Authors

Reviewer 4 Report

Dear authors,

I put my comments on the text attached!

Good luck!

Author Response

Dear respected reviewer 

Hope you are doing well

Thank you very much. We are very honored and grateful

With appreciation 

Sincerely yours

Authors

Round 3

Reviewer 1 Report

I would like to argue once again. There is a significant problem with the story you want to convince anyone to read. It also definitely refers to “so what” question. I suppose you have plenty of ideas justifying your research and some difficulties to choose the one to follow. See that in one place you state that your paper “examines the decision-making process to learn why tourists cancel their travel plans” then you want to investigate “what influence their travel risk perception under a [specific] scenario”, finally you investigates “whether the effects of perceptions of tourism have an impact on economic and financial development based on [some] factors”. This really make difficult anyone to understand what you want to achieve. And this is probably the reason you received some contradictory opinions from different reviewers. Moreover, there are no clear linkages between the problem you want to define, literature review (with a very limited content about perception of tourism), results and conclusions.

Just to clarify, I didn't say that the results section from the perspective of econometrics is wrong. Everything is ok with that. However, it is not clear what is the story you presented in that section. It is not clear where you want to go with that paper. See my comments above. From the econometric perspective – everything is fine, from the analytical point of view – it is no clear where you want to go. You present the estimations, but you do not analyze and interpret the results.

Copied maps are unfortunately unacceptable. Try to use some GIS software to elaborate that and deliver own maps. There are some specific requirements of editing maps. Try to find some support in the field of cartography.

Author Response

Dear Reviewer 

We have improved all according to the suggestions given by you. 

Thank you for your support and much appreciated.

With gratitude and appreciation 

Authors.

Reviewer 4 Report

Dear authors,

Thank you so much for this improved version of your article.

Recommendations for improving this research have been largely implemented.

However, I suggested to the authors that the literature review is very large and lacks a clear structure.

Also, the authors did not take into account the recommendation regarding the writing of the results from the tables, but also from the text. The authors use "dot" or "comma" without following a write rule. Here you should see the author's guide and English-style writing.

That's all!

Best wishes!

Author Response

Dear Reviewer 

We have improved all according to the suggestions given by you.

Thank you once again for your support. Much appreciated.

Authors 
